# Subcortical Structures in Demented Schizophrenia Patients: A Comparative Study

**DOI:** 10.3390/biomedicines11010233

**Published:** 2023-01-16

**Authors:** Juan Rivas, Santiago Gutierrez-Gomez, Juliana Villanueva-Congote, Jose Libreros, Joan Albert Camprodon, María Trujillo

**Affiliations:** 1Department of Psychiatry, Fundación Valle del Lili Cra. 98 # 18-49, Cali 760032, Colombia; 2Department of Psychiatry, Universidad ICESI, Cali 760031, Colombia; 3Department of Psychiatry, Universidad del Valle, Cali 760043, Colombia; 4Hospital Departamental Psiquiátrico, Universitario del Valle, Cali 760035, Colombia; 5Centre for Research and Training in Neurosurgery (CIEN), Bogotá 110411, Colombia; 6Neurosurgery Department, Universidad de Nuestra Señora del Rosario, Bogotá 111711, Colombia; 7Research Office, Hospital Universitario San Ignacio, Bogotá 110231, Colombia; 8School of Systems and Computing Engineering, Universidad del Valle, Cali 760032, Colombia; 9User-Centric Analysis of Multimedia Data Group, Technische Universität Ilmenau, 98693 Ilmenau, Germany; 10Division of Neuropsychiatry, Massachusetts General Hospital, Harvard Medical School, Boston, MA 02129, USA

**Keywords:** schizophrenia, dementia, aged, hippocampus, amygdala, thalamus, neuropsychological tests

## Abstract

There are few studies on dementia and schizophrenia in older patients looking for structural differences. This paper aims to describe relation between cognitive performance and brain volumes in older schizophrenia patients. Twenty schizophrenic outpatients —10 without-dementia (SND), 10 with dementia (SD)— and fifteen healthy individuals —as the control group (CG)—, older than 50, were selected. Neuropsychological tests were used to examine cognitive domains. Brain volumes were calculated with magnetic resonance images. Cognitive performance was significantly better in CG than in schizophrenics. Cognitive performance was worst in SD than SND, except in semantic memory and visual attention. Hippocampal volumes showed significant differences between SD and CG, with predominance on the right side. Left thalamic volume was smaller in SD group than in SND. Structural differences were found in the hippocampus, amygdala, and thalamus; more evident in the amygdala and thalamus, which were mainly related to dementia. In conclusion, cognitive performance and structural changes allowed us to differentiate between schizophrenia patients and CG, with changes being more pronounced in SD than in SND. When comparing SND with SD, the functional alterations largely coincide, although sometimes in the opposite direction. Moreover, volume lost in the hippocampus, amygdala, and thalamus may be related to the possibility to develop dementia in schizophrenic patients.

## 1. Introduction

Schizophrenia is a severe mental disorder whose aetiology includes anatomical, genetic, and environmental factors, with significant effects on patients and society [1]. Cognitive disorders are a central finding in the disease [2], with some patients showing performance that is one standard deviation below the general population in cognitive testing. Nevertheless, in a significant number of patients, cognitive impairment is not described [3], suggesting that the problem is not clear yet.

There are different trajectories in the ageing processes of patients with schizophrenia compared to people with dementia and healthy individuals [4]. People with schizophrenia could have a higher risk of dementia than general population [4], but there is no consensus about these findings [5]. The risk for developing dementia is twice as high in patients with schizophrenia compared to healthy people, especially in those younger than 65-years [6,7,8,9,10].

Furthermore, the risk for dementia in late-onset and very late-onset schizophrenia can rise by 400% [11]. Inconsistencies among reports could be explained by multiple variables such as clinical status [3], severity of symptoms [12], number of hospitalizations [13], severity of negative symptoms [14], environmental factors [7], and somatic comorbidities [15]. Deficits in visuospatial orientation, memory, and attention have been described, even during the early stages of the disease [16]. However, there is no consensus on if abnormalities occur before patients are diagnosed with schizophrenia or if there is a cognitive decline that occurs across time [12]. 

In schizophrenia, the hippocampus is the structure that has been shown to have the largest volume loss [17,18,19]. Alterations are more frequently found on the left side, especially in the anterior hippocampus, CA1, and subiculum [20]. Although neuronal loss is not evident [21], a 4% bilateral hippocampal volume reduction can be observed regardless of the disease’s duration, age of onset, and medication. Patients with early-onset schizophrenia also show amygdala volume reduction on the left side, [17,19], in the basal nuclei, anterior amygdaloid area, paralaminar nuclei, and lateral nuclei [22]. Regarding the thalamus, the most frequent nuclei affected in schizophrenia are the dorsomedial (DM), ventral anterior (VA), and pulvinar, in which a reduction in the number and volume of neurons is evident [23]. Although the loss in thalamic volume seems nondependent on the chronicity of the disease [24], thalamic atrophy during the transition to psychosis in patients with poor prognosis has been reported [25].

Few studies focus on the risk of dementia in older schizophrenic patients and its relations with structural changes in the hippocampus, amygdala, and thalamus. Most of the studies evaluate young people in the first psychotic episode and they often ignore the effect of age on cognitive decline and structural changes [26,27,28,29,30]. Moreover, studies involving patients older than 50-years-old do not control for the risk factors inherent to the disease and those from the ageing process [10,26,31]. Additionally, there is scarce literature focused on the relations between schizophrenia and dementia in Latin American populations [32].

An understanding of structural deficits and cognitive impairment would allow us to develop biomarkers for older SZ and SZ dementia patients. The current study aims to investigate differences in cognitive performance, hippocampal, amygdala, and thalamic volume, and relations between cognitive functioning and structural changes in a group of schizophrenic patients older than 50 years.

## 2. Materials and Methods

### 2.1. Participants

Using a non-probabilistic sampling, we selected 20 outpatients and 15 healthy individuals, older than 50 years, from two hospitals in Cali, Colombia: Hospital Departamental Psiquiátrico Universitario del Valle (HDPUV) and Fundación Valle del Lili. We defined the groups as follows: 10 patients with schizophrenia without dementia (SND), 10 patients with the previous diagnosis of schizophrenia and recently diagnosed with dementia (SD), and 15 healthy subjects that were taken as the control group (CG). The symptoms of patients with schizophrenia started before 30 years old. The IRB of both hospitals approved the study and the Declaration of Helsinki was followed. All individuals signed informed consent before being included in the study.

To select subjects, we reviewed the charts of schizophrenic patients. In their clinical reports they were diagnosed at some point and were treated as such. JR reviewed every chart and two independent psychiatrists certified the diagnosis using DSM V criteria for schizophrenia. The Positive and Negative Syndrome Scale (PANSS) was used to evaluate the severity of positive and negative symptoms. Patients were on antipsychotics as part of their treatment. 

Dementia diagnosis was performed using the DSM V criteria for major neurocognitive disorders. Dementia patients belong to the neuropsychiatric clinic in HDPUV, and were evaluated by a neuropsychiatrist, neuropsychologist, and neuroradiologist. If there were no coincidences in diagnosis, an additional psychiatrist, with experience in dementia, was asked to perform the definitive diagnosis.

Exclusion criteria were neurological diseases such as epilepsy, stroke, traumatic brain injury, CNS infection, and brain tumours.

The null hypothesis was that there are no structural and functional differences when comparing the three groups. Raw data were previously published [33].

### 2.2. Cognitive Evaluation

Two neuropsychologists carried out screening, i.e., cognitive evaluations. They used MMSE and Addenbrooke’s Cognitive Examination (ACE). Additionally, the CDR and the Hachinski Ischemic Score (HIS) [34] were applied to confirm the presence of dementia. Finally, all subjects underwent the Yesavage Geriatric Depression Scale [35].

Neuropsychological tests included the Hopkins Verbal Learning Test (HVLT) [36], the Rey Complex Figure Test, [37], and the Free and Cued Selective Reminding Test (FCSRT) [38] for analysing memory. The Hopkins Verbal Learning Test and the FCSRT were applied for learning and memory capacity at the auditory–verbal level, and the Rey Complex Figure Test (RCFT) was used for the encoding and evocation of graphic visual material. The Boston Naming Test (BNT) was used for linguistic function [39].

The phonological fluency test (Letter F and S), [40] the semantic fluency test (animal creep), and the digit span test (DST) were used to assess prefrontal cortex functioning. Finally, the semantic fluency test (animal fluency) was also applied to assess mental flexibility and categorization.

### 2.3. Structural Reconstruction

All patients underwent MRI. Images were taken at the Fundación Valle del Lili using a 1.5 Tesla Siemens Avanto resonator, using the following parameters: repetition time (ms) 8000, echo time (ms) 99, inversion time (ms) 2371.2, flip angle 150, layer thickness 5 mm, space between layers 6 mm, voxel size 1 mm^3^, in axis x = 256, y = 256, and z = 0.898438.

### 2.4. Cortical Surface-Based Analysis

#### 2.4.1. FreeSurfer 6

Cortical reconstruction and volumetric segmentation were performed using the FreeSurfer 6 image analysis suite, which is available at http://surfer.nmr.mgh.harvard.edu/, accessed on 1 July 2019. The technical details of these procedures are described in several publications [41,42,43,44,45].

#### 2.4.2. Cortical and Volumetric Segmentation

This process includes: (1) motion correction and averaging of multiple volumetric T1 weighted images (when more than one is available), (2) removal of non-brain tissue using a hybrid watershed/surface deformation procedure [46], (3) automated Talairach transformation, (4) segmentation of subcortical white matter and deep grey matter volumetric structures—including the hippocampus, amygdala, caudate, putamen, ventricles—[42], (5) intensity normalization [47], (6) tessellation of the grey matter, the white matter boundary, automated topology correction [48], and (7) surface deformation following intensity gradients to optimally place the grey/white and grey/cerebrospinal fluid borders at the location with the most significant shift in intensity. This defines the transition to the other tissue class [41,49].

MRI images were automatically processed with the longitudinal stream in FreeSurfer 6 to extract reliable volume and thickness estimates [50]. Specifically, an unbiased within-subject template space image is created using robust, consistent inverse registration [50]. Several processing steps, such as skull stripping, Talairach transforms, MNI atlas registration, and spherical surface maps parcellations were initialised with shared information from the within-subject template for increased reliability and statistical power (Reuter et al., 2012).

For segmentation and parcellation of hippocampus and amygdala, we used the developing version at (https://surfer.nmr.mgh.harvard.edu/fswiki/HippocampalSubfields, accessed on 1 July 2019) with specific nomenclature [51]. For the thalamus, we used the developing version at (http://freesurfer.net/fswiki/ThalamicNuclei, accessed on 1 July 2019) with specific nomenclature [52].

#### 2.4.3. Statistical Analysis

Once data were obtained, we ran a normal distribution test on it and discovered that the data do not follow a normal distribution. Thus, we used non-parametric statistical tests. Differences between the three groups were tested using the Kruskal–Wallis and Mann–Whitney U tests, with neuropsychological scores and volumes of the thalamus, amygdala, and hippocampus. The null hypothesis was rejected with *p* ≤ 0.05. Regression models were used to explore relations between brain volumes and demographics data, along with cognitive performance, since there is no way to calculate correlation between three or more variables. The three demographic variables—age, years of schooling, mental disease duration—and neuropsychological scores were set as independent variables and a selected variable—volume of thalamus, amygdala, or hippocampus—was set as the dependent variable. The regression models were evaluated and selected based on the coefficient of determination, set at R^2^ ≥ 0.8. Once a regression model had R^2^ ≥ 0.8, the contribution of the independent variables to the model was assessed by the significance value of the coefficient of a variable—contributing to the model when *p* ≤ 0.05. We used the package Stata 16.0 (Stata Statistical Software StataCorp 2019) for the Kruskal–Wallis and the Mann–Whitney U tests, defining significance tests as *p* ≤ 0.05. Additionally, for each regression model, Spearman rank higher order correlation was used to corroborate pair relations between volume and demographic variables, as well as a neuropsychological test. We used ρ ≥ 0.8 for determining significance of the Spearman correlation.

Briefly, we used regression models for relating structure volumes with demographic and cognitive performances, then we corroborated those relations using Spearman correlations.

## 3. Results

We evaluated 35 individuals, 19 were women (54.2%). The median ages were 69.5 years for SD, 58 for SND, and 60 for CG. The Kruskal–Wallis tests showed significant differences between ages when comparing the three groups. Moreover, the Mann–Whitney U tests showed significant differences between ages of CG-SD and SD-SND (*p* 0.006 and 0.003 respectively), but not when comparing CG to SND (*p* = 0.18). The schizophrenic groups had a lower educational level than CG, without finding significant differences between them (*p* = 0.21). There was a significant difference in mental illness duration (*p* = 0.0042), with a median of 41.3 years in the SD group and 26.9 in the SND. Table 1 shows the order statistics of demographic variables, along with the obtained Mann–Whitney U tests *p*-values of comparison between groups.

In the PANSS scale, there were significant differences between the groups in negative symptoms and general psychopathology, with better performance in SND. Patients had difficulty in abstract thinking (*p* = 0.015), stereotyped thinking (*p* = 0.039), anxiety (*p* = 0.034), uncooperativeness (*p* = 0.009), disorientation (*p* = 0.001), poor attention (*p* = 0.001), lack of judgment and insight (*p* = 0.001), disturbance of volition (*p* = 0.008), poor impulse control (*p* = 0.016), and in the total general psychopathology scale (*p* = 0.001). There were no significant differences in positive symptoms. Order statistics of test results are detailed in Appendix A (Table A1).

Patients were on antipsychotics as part of their treatment. Throughout the course of their illness, they received both typical and atypical antipsychotics. Since SD were older, obviously they were exposed for longer periods of time to pharmacological management than SND. However, the registers are of formulation, but there is no certainty of adherence to it.

Table 2 summarises the quartiles of score of the different neuropsychological tests for the functional analysis and the obtained *p*-values of Mann–Whitey U tests. There were significant differences in overall cognitive condition between the three groups. We found that the CG performed within normal limits, while patients with schizophrenia from both groups had an inferior performance, with evident alterations in the different domains. Moreover, we observed a greater severity in SD than in SND in all tests, except the semantic memory and visual attention tests.

### 3.1. Structural Analysis

Kuskal–Wallis tests were used for examining the differences between the three groups for the set of variables. Mann–Whitney U tests results—with *p* ≤ 0.05—are illustrated in Figure 1, Figure 2 and Figure 3. The green line indicates differences between CG and SD, the red line indicates differences between SD and SND, and the blue line indicates differences between CG and SND. Moreover, the *y*-axis represents the magnitude of volumes and the *x*-axis corresponds to the segmented structures. 

#### 3.1.1. Hippocampus

Hippocampal volumes did not show significant differences when comparing SND with CG using Mann–Whitney U (See Figure 1). When we compared SD with CG, there were significant differences in every segment, with higher compromises on the right side. On the left side, SD had less volume than CG at the granular layer in the head of the dentate gyrus, head of CA4, fimbria, head of CA3, body, and head of the hippocampus. On the right side, findings reached significance on the head and body presubiculum, fimbria, head of CA3, and the entire hippocampus’s body. Medians are presented in Appendix B (Table A2).

**Figure 1 biomedicines-11-00233-f001:**
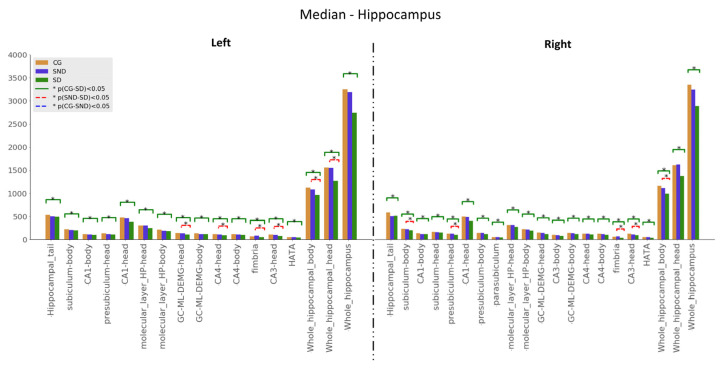
Comparison of medians, $H_{0}:\mu_{i}=\mu_{j}$, with Mann–Whitney U test, with * *p* ≤ 0.05, for hippocampus structures.

#### 3.1.2. Thalamus

In the thalamus analysis, we used the non-motor or sensory relay nuclei: the anterior (AV), the dorsal medial (MDI), and the pulvinar. The latter was subdivided into subnuclei, namely anterior (PuA), lateral (Pul), medial (PuM), and posterior. When comparing SD with CG, all the volumes were significantly higher in CG with statistically significant results expected in the right PuA (Figure 2). Comparisons between SND and SD showed statistically significant results in the left MD nucleus (both parvocellular and magnocellular divisions) and the left thalamus’s whole volume, higher in SND than SD. In all structures under investigation, volumes were higher in CG > SND > SD, except in the right thalamic whole volume, in which SND held the higher measurements followed by CG and SD. Medians are presented in Appendix B (Table A3).

**Figure 2 biomedicines-11-00233-f002:**
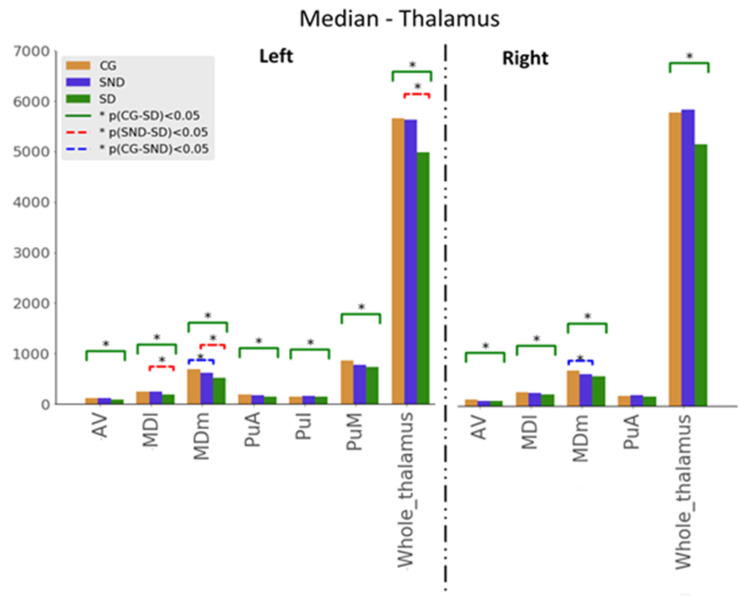
Comparison of medians, $H_{0}:\mu_{i}=\mu_{j}$, with the Mann–Whitney U test, with * *p* ≤ 0.05, for thalamus structures (AV: Anteroventral, MDm: Mediodorsal medial, MDl: mediodorsal lateral, PuA: Anterior pulvinar, PuI: Inferior pulvinar, PuM: Medial pulvinar).

#### 3.1.3. Amygdala

Amygdala analyses were based on the anatomical nuclear division, as shown in Figure 3. As in all the previous analyses, comparisons between CG and SD showed significant differences in all the studied structures, with SD having the lowest volumes. Regarding SND vs. SD comparisons, statistical significance was reached for the left whole amygdala volume and in the right basal nucleus, with the lowest volumes in SD. In all structures, volumes were ordered as follows: CG > SND > SD. Overall, the left amygdala showed lower volumes than the right amygdala. Medians are presented in Appendix B (Table A4).

**Figure 3 biomedicines-11-00233-f003:**
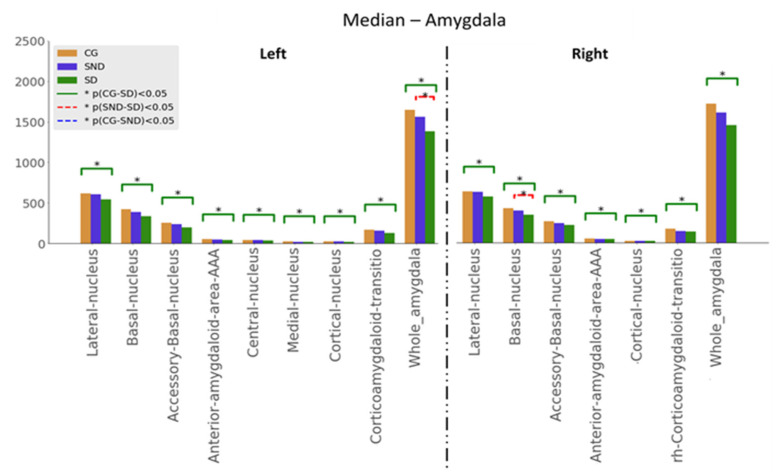
Comparison of medians, $H_{0}:\mu_{i}=\mu_{j}$, with the Mann–Whitney U test, with * *p* ≤ 0.05, for amygdala structures.

### 3.2. Structure vs. Function Analysis

Linear regression models were calculated to establish relations between the volumetric variables—each as the independent variable—and demographic variables, along with neuropsychological test scores as dependent variables. If R^2^ ≥ 0.8, the regression model is considered representative of data and selected as a good model for representing relations between volumetric and significant dependent variables.

Then, Spearman correlations were calculated for corroborating possible relations between structures and functions. Figure 4, Figure 5 and Figure 6 show regression models with R^2^ ≥ 0.8, where black boxes correspond to the structures, blue lines indicate positive relations, and red lines, negative ones with cognitive performance and demographic data that each variable coefficient contributed to the model (*p* ≤ 0.05). Additionally, Spearman correlation results are shown in Figure 4, Figure 5 and Figure 6, where grey points mean that there is no correlation.

The regression models for the CG did not yield R^2^ ≥ 0.8. The larger R^2^ values on the right side were 0.36 in the hippocampus, 0.56 in the thalamus, and 0.31 in the amygdala, while on the left side, they were 0.62 in the hippocampus, 0.58 in the thalamus, and 0.23 in the amygdala.

#### 3.2.1. Hippocampus

In SND patients, the left head presubiculum showed a positive correlation with schooling and a negative correlation with animal fluency and HVLT delayed recall. Similarly, the subiculum body showed a positive correlation with schooling and a negative correlation with animal fluency. The right side analyses showed relations in the head presubiculum and subiculum, in such a way, that the presubiculum was positively related to schooling and negatively to HVLT delayed recall, whereas the subiculum head had a positive relation with schooling and disease years, while showing a negative relation to HVLT delayed recall test (Figure 4).

On the other hand, SD patients showed relations in the CA3 segment of the hippocampal head. On the right side, positive relations were found regarding age, schooling, Rey complex figure copy, FCSFT free recall score, and a negative relation regarding years of the disease. On the left side, positive relations were found with age, schooling RCFC and delayed recall, and FCSRT free recall score; negative relations were related to years of diagnosis (Figure 4).

**Figure 4 biomedicines-11-00233-f004:**
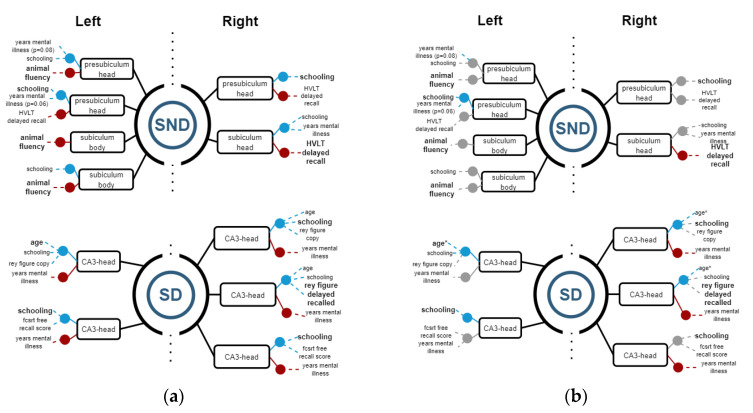
(**a**) Regression models with R^2^ ≥ 0.8, indicating the structure variables in a black rectangle aligned with blue (+) and red (−) points for the significant independent variables. (**b**) Spearman correlations with ρ ≥ 0.8 for corroborating relations between the structures and demographic and cognitive variables. Grey points means that there is no correlation.

In SND, only HVLT showed a negative correlation with the right subiculum head, while in the left side, only schooling had a positive one. In the SD group, there were more correlations: CA3 head on the right side had positive correlation with age, and negative ones with years of mental disease. On the left side, there were positive correlations between head of presubiculum and schooling, and a positive correlation between CA3 head, age, and schooling.

#### 3.2.2. Thalamus

Among SND patients, regression models showed relations on the left side similar to their SD counterparts. On the left side, the MDl had a positive relation with HVLT total recall and the MDl had a positive relation with HVLT delayed recall. The Pul volume can be estimated based on a positive relation with the Rey figure delayed recall and negative relation with age and schooling. The magnocellular portion of the AV nucleus was positively related to schooling and animal fluency. The VLA had positive results with age, years of mental disease, and schooling, while showing a negative relation with cued recall score. The whole left thalamus had positively related results in schooling, Boston naming tests, and years of the disease. On the right side thalamus in the SND group, the MDl nucleus had positive relations with schooling and years of disease, while showing negative relations with Rey’s figure copy. The Pul had a positive correlation with years of the disease. The PuM nucleus had a positive relation with the Boston naming test and years of mental disease (Figure 5).

In SD patients, relations were found on the right side. The right AV nucleus was negatively related to digit span, age, schooling, and years of diagnosis. Additionally, the AV nucleus was positively related to years of disease and the Boston naming test. On the left side, MGN had a positive relation with HVLT delayed recall (Figure 5).

**Figure 5 biomedicines-11-00233-f005:**
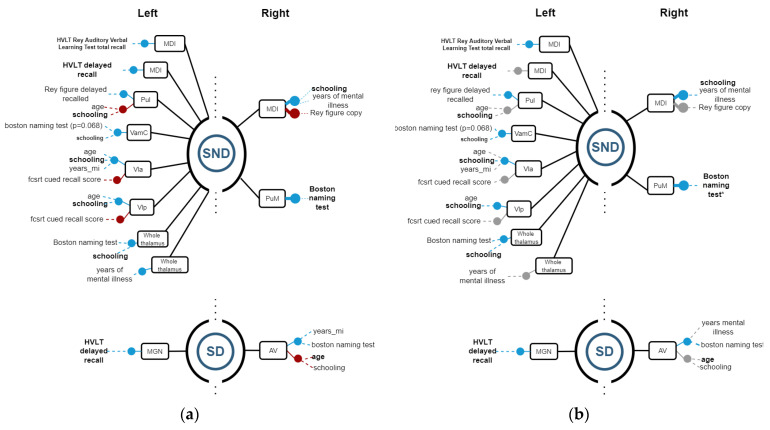
(**a**) Regression models with R^2^ ≥ 0.8, indicating the structure variables in a black rectangle aligned with blue (+) and red (−) points for the significant independent variables. The graph on (**b**) indicates Spearman correlations with ρ ≥ 0.8 for corroborating relations between the structures and demographic and cognitive variables. Grey points mean that there is no correlation.

Regarding Spearman correlations, we found positive correlations on the right side between MCI and years of mental disease, along with PuM and Boston naming in SND. On the left side, there were positive correlations between MCI and total recall of HVLT Rey’s words, Pul and Rey´s figure delayed recall, VamC and schooling, Vla and schooling, Vlp and schooling, and the whole thalamus and Boston naming. In SD, on the right side, AV was correlated with years of mental disease, while on the left side, MGN had a positive correlation with hvlt delayed recall.

#### 3.2.3. Amygdala

In SND patients, all statistically significant results between structures and functions were found on the right side. All the right segments had a strong positive relation to schooling, while most of them showed negative correlations with the HVLT delayed recall test except the medial nucleus, where a negative relation was found on the FCRST Iden Test. The right paralaminar nucleus also showed a positive correlation with age but to a lesser extent with schooling. Regarding SD patients, only negative relations were found, and only on the left side. The medial and the cortical nucleus had negative relations with age (mainly), schooling, and HVLT words of Rey total Recall tests (Figure 6).

By the correlation analysis, we corroborated, in SND on the right side, that basal nuclei, medial nuclei, and whole amygdala had a positive correlation with schooling. In SD, significant correlations were not possible to corroborate on the left side.

**Figure 6 biomedicines-11-00233-f006:**
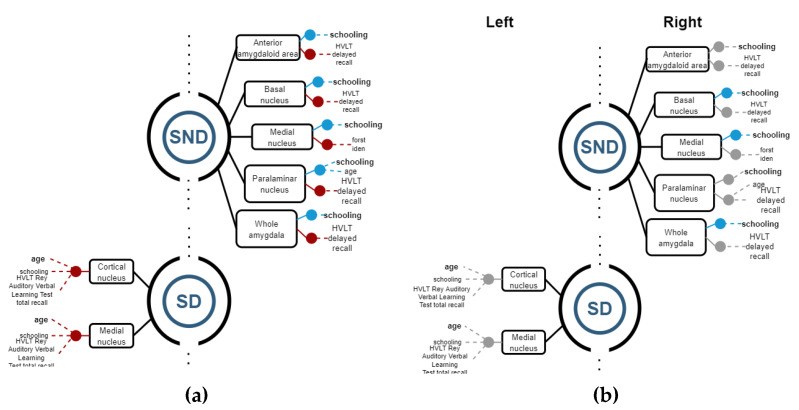
(**a**) Regression models with R^2^ ≥ 0.8, indicating the structure variables in a black rectangle aligned with blue (+) and red (−) points for the significant independent variables. (**b**) Spearman correlations with ρ ≥ 0.8 for corroborating relations among the structures and demographic and cognitive variables. Grey points means that there is no correlation.

## 4. Discussion

We evaluated 35 individuals to compare cognitive performance and brain volumes. Cognitive performance was significantly lower in schizophrenia patients (SD and SND) than in CG. Structurally, we found statistically significant differences among schizophrenia patients when compared to CG in the hippocampus.

The cause of different performance in cognitive evaluation remains unclear. Demographic and other non-schizophrenia-related factors such as age did not fully explain this finding. If age was indeed a factor, SND patient performance would have been similar to CG and not to SD. Both groups of patients with schizophrenia presented failures in the same cognitive domains. While overall cognitive impairment was greatest in SD over SND, visuo-graphic memory and executive system function impairment was consistent between the two schizophrenia groups.

Although it is well known that schizophrenia causes a degree of cognitive impairment and that cognitive deficits in these patients are established early in the disease, which makes them distinguishable from the healthy population. The structural basis for its onset appears to be different from that found in AD or frontotemporal dementia (FTD) [8,25,31,53,54,55,56,57,58,59,60,61,62,63,64,65,66,67,68,69,70,71].

Deficits in left CA1, subiculum, and DG were shown in schizophrenia [20,72], while in AD, there is a compromise of CA1 and the subiculum, preserving CA3 and DG [73]. There was a gradual reduction in the volume of CA3 in the SD in our sample, directly related to visual memory. The latter could imply a coexistence of two pathologies, explained by factors such as the sample size, the presence of dementia other than AD, or the chronicity of schizophrenia which further deteriorates these structures. An analogous situation occurred with the head of the left subiculum, which had a negative relation with audio-verbal memory in SD, but not in SND. This alteration could be caused by the dementia process and not for schizophrenia; therefore, CA3 volume and its associated function may also constitute markers of cognitive impairment in schizophrenic patients.

In the thalamus, the magnocellular portion of the left MD nucleus was significantly smaller in SD than SND and CG and SND than CG, suggesting that schizophrenia could be the preponderant factor for reducing the size of this nucleus. On the right side, the differences are significant when comparing the groups with schizophrenia to the CG, but not when comparing SD with SND. This thalamic alteration could be a specific compromise for the dementia process, and not influenced by the schizophrenic process.

The thalamic nuclei associated with cognitive functions are the DM, AV, and pulvinar [74,75,76]. Consequently, alterations in these structures have been associated with cognitive deficits, such as attentional, executive, and language failures in schizophrenia [77,78]. Thalamic alterations in our sample were evident in the nuclei related to cognition on the left side and related, although secondarily, to short-term memory processes.

Regarding the amygdala, there were no significant differences between CG and SND, and a few regions with significant differences between SD and SND, which led us to think that the loss of volume in this structure is not associated with schizophrenia and that the associated mental symptoms may be a consequence of its atrophy. This may imply that, unlike AD, where the main alterations occur in the hippocampus, in schizophrenia, a predominant factor in the genesis of dementia is the atrophy of specific nuclei of the amygdala and thalamus. A structural marker of risk or early diagnosis of dementia in schizophrenic patients could therefore be available, since there is evidence of the modulation that the amygdala exerts on the hippocampus for episodic memory [79,80,81].

The Spearman correlation and the linear regression did not always show relations between structural differences and cognitive function. This implies that the cognitive function does not rely only on volumetric measurements of brain structures, but on the connectome integrity in terms of functional analysis. It will be necessary to consider that volume does not necessarily relate to function or that there are compensatory brain mechanisms against the atrophy of some of its structures.

The lack of coincidence between the findings previously reported in the literature and our results can be explained by the analysis and segmentation methods, and the age group. Segmentation of the brain is a complex procedure and only through sophisticated image processing methods is it possible to separate some structures from others. The used FreeSurfer version 6 allows a more precise segmentation of small structures, such as the specific thalamic nuclei and the amygdala, and differentiation of segments of the hippocampus.

## 5. Conclusions

Cognitive performance and structural changes allowed us to differentiate between schizophrenia patients and CG. Changes in both domains were more evident in SD than in SND, suggesting that schizophrenia may be a risk factor for developing dementia symptoms and that may be explained by changes in the hippocampus, thalamus, and amygdala.

When comparing SND with SD, the cognitive alterations coincide, although sometimes in the opposite direction. This could be explained in several ways: (a) an insufficient sample size, (b) the impact of demographic variables, especially age, and (c) the duration of the mental illness. Other structures than expected for the underlying pathology appear to be compromised. If this is the case, these differentiated structures may become markers of deterioration for patients with schizophrenia and without dementia.

This study has several strengths and limitations. Our research has various novelties. First, we selected a cohort of patients older than 50, which has not been frequently reported in the literature. Second, we used image segmentation software (FreeSurfer-6), which allowed us to obtain more details than previously published studies. Additional, there are few studies on the relation between schizophrenia and dementia in Latin America [32]. Finally, we did a structure–function analysis looking for possible relations between them.

Among the limitations, the first one is the sample size, a drawback that arises from several facts. The prevalence of dementia in schizophrenic patients is not established; therefore, an exploratory study was conducted. The obtained results cannot be extrapolated to patients universally. Moreover, there were economic limitations that prevented a larger sample size and deepened possible causal factors of dementia symptoms. In the findings, the age bias shown by the SD group implies that age may be a confounding factor. This inconvenience arose due to the limited number of patients that were enrolled.

## Figures and Tables

**Table 1 biomedicines-11-00233-t001:** Demographic variables. CG: Control Group; SND: Schizophrenia without Dementia; SD: Schizophrenia with Dementia.

	CG Q2 (Q1–Q3)	SND Q2 (Q1–Q3)	SD Q2 (Q1–Q3)	Mann–Whitney *p*-Value		
				CG vs. SND	CG vs. SD	SD vs. SND
Age (years)	60.0 (56.0–64.0)	58.0 (52.0–59.8)	69.5 (66.5–72.5)	0.1810	0.0060	0.0030
Schooling: completed years	15.6 (11.0–22.0)	8.9 (5.0–14.0)	6.8 (0–15.0)	0.0001	0.0001	0.2122
Years of schizophrenia	-	26.9 (15.0–45.0)	41.3 (27.0–55.0)	-	-	0.0042

**Table 2 biomedicines-11-00233-t002:** Quartiles of neuropsychological scores. CG: Control Group; SND: Schizophrenia without Dementia; SD: Schizophrenia with Dementia.

	CG Q2 (Q1–Q3)	SND Q2 (Q1–Q3)	SD Q2 (Q1–Q3)	Mann–Whitney *p*-Value		
				CG vs. SND	CG vs. SD	SD vs. SND
LetterF	6 (4–7)	4 (3.8–5.3)	0 (0–2.3)	0.039	0.001	0.001
Animal Fluency	7 (6–7)	3 (1.8–4.3)	0 (0–2)	0.001	0.001	0.005
LetterS	6 (4–7)	3 (0–4)	0 (0–0.3)	0.002	0.001	0.034
HVLT Rey words-Total Recall	105 (87–116)	56.5 (41–75)	25.5 (14.3–42.3)	0.001	0.001	0.009
HVLT-Delayed Recall	11 (9–14)	4.5 (3.8–7.3)	0 (0–3)	0.001	0.001	0.007
Rey Figure-Copy	35 (32–36)	25.5 (18.4–32)	7 (0–14)	0.003	0.001	0.025
Rey Figure-Immediate Recall	18 (14.5–28)	7.5 (0–11)	0 (0–3.8)	0.001	0.001	0.122
Rey Figure-Delayed Recalled	18 (14–22.5)	7.5 (0.4–13.5)	0 (0–2)	0.001	0.001	0.024
Digit Span	5 (4–7)	4 (2.8–4)	2.5 (1.5–3.0)	0.005	0.001	0.012
Boston Naming Test	19 (19–20)	8 (0–19.3)	11.5 (0–15)	0.018	0.001	0.617
FCSRT-IDEN	16 (16–16)	15 (13.8–16)	12.5 (7.5–15.3)	0.001	0.001	0.077
FCSRT-Free Recall score	35 (31–40)	18 (12.8–26.5)	8.5 (0–13.5)	0.004	0.001	0.022
FCSRT-Cued Recall score	43 (40–46)	24 (17.3–30.8)	7 (0–21.3)	0.001	0.001	0.033
FCSRT-Total recall score	73 (55–84)	46.5 (37.5–55.5)	18 (0–4)	0.017	0.001	0.015

## Data Availability

All the original data used in this research are available at https://zenodo.org/record/3901876, accessed on 20 June 2020 [82].

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
