# Peer review of "Subcortical Structures in Demented Schizophrenia Patients: A Comparative Study"

_biomedicines, 2023, doi:10.3390/biomedicines11010233_

Round 1

Reviewer 1 Report

The authors compared the performance of different cognitive domains and key brain volumes calculated from magnetic resonance images between 10 without-dementia (SND), 10 with dementia (SD), and 15 healthy controls (CG) and also investigated the relations between cognitive performance and brain volumes. The paper has the potential to contribute to the existing scientific literature on the risks for older schizophrenia patients developing dementia and its relations with structural changes in the hippocampus, amygdala, and thalamus.  I only have a few comments to further improve the quality of the authors’ paper. I have outlined these issues below:

1.

In DSM 5 major Neurocognitive Disorders, the primary recognized neurocognitive disorders include: Alzheimer’s disease, frontotemporal degeneration, Huntington’s disease, Lewy body disease, traumatic brain injury (TBI), Parkinson’s disease, prion disease, such as Creutzfeldt-Jakob disease or Bovine Spongiform Encephalopathy (“mad cow disease”), dementia/neurocognitive issues due to HIV infection, and vascular dementia.

The readers may wonder the individual impression of the possible etiology of dementia among the 10 patients with dementia (SD). For patients with AD, was the ATN system used?  Any results of amyloid or tau PET study?

2.  Row data was previously published [33].

A typo, “row” should be “raw”.

3. The readers may wonder the rationale of Neuropsychological tests  selected. Was there any cognitive battery for schizophrenia and dementia suggested by the specialist or research consensus? If any, please provide references.

4. page 3, line 21.

The phonological fluency test (Letter F and S), [40] the semantic fluency test (animal creep), and the digit span test (DST) were used to assess the prefrontal cortex functioning. Finally, the semantic fluency test (animal fluency) was applied to assess mental flexibility and categorization.

The readers may wonder the difference between the semantic fluency test (animal creep) versus the semantic fluency test (animal fluency). Please brief it.

Also, for the digit span test, was it forward or backward?

5. The method is well-written.  The readers may wonder more clear the description regarding the voxel size. Please specify it.  

6.  Figure 2  “MDm” , the abbreviation of what structure?

7.  In the legend of figure 4.  There is a typo “variabl.es.”

 8. For the tile,  "schizophrenic". It would be better to replace the term "schizophrenic" with "schizophrenia".    

9. 

upper case or lower case of Hopkins verbal learning test (HVLT) can be consistent throughout the manuscript, tables and figures.

10. In figures 4-6,  what did grey points mean?  what did the asterisk footnote* mean?  why did the authors list the variable with the p value greater than 0.05 ?  These can be mentioned in the figure legend. 

11.  The discussion is well-written.

page 11, line 28. 

this opens the possibility that schizophrenia itself could be an alternate pathway to dementia.

The authors may briefly discuss some potential molecular mechanisms for the alternate pathway from schizophrenia to dementia. 

In the reviewer’s opinion, the above-mentioned issues need to be addressed by the authors.

Reviewer 2 Report

The article "Subcortical structures in demented schizophrenic patients: a comparative study" is a pilot study using a 1.5T MR scanner to investigate whether patients with a schizophrenia spectrum diagnosis (SSD) and dementia show reduced volume in three brain regions; HPC; Amygdala and thalamus, compared to patients with SSD that have no dementia as well as a control group.

The paper is in many respects untypical as it lacks
a) detailed description of the patients, e.g. table of medication, exact DSM V diagnosis (e.g. psychotic or not, language deficits)
b) PANSS scores
c) for the control group details are not provided either, but they are referred as patients, so did they had another disorder or mix of disorders?

methods: R2 (R square) of .8 means that a 80% of the variance is explained. That is more than untypical to find for regressions between brain volume data (or any brain data) and demographics or any kind of cognitive performance. Do you mean .08?

also note that Spearman's r is an effect size, not a cut-off criterion for significance. Sample size and effect size influence whether the p value is below or above the 5% criterion

analysis: since you have left/right and subregions you have to control for multiple testing. In FSR or SPM one can use the false discovery rate or other methods, also stricter criteria like p < .001 are common.

results: please provide brain images with color codes, i.e. comparing SND with SD or SD with CG and correlation, typical images show brain regions that are "redish" (difference in favour of group 1) or blueish (difference in favour of group 2).

your raw data (btw it is raw data not row data) is a csv file but has n=70, of which n=30 seem to be control, n=18 SND and n=22 SD

please explain why you in this manuscript only report data from n=10 SD, n=10 SND and n=15 C

The statistical analysis is also untypical, you have 3 groups and brain regions, and your question is whether the groups differ, hence this is a GLM or repeated measure ANOVA with between-factor the 3 groups, and within factor the brain region of focus (subnuclei and left vs right)

then you can add as co-variate age, gender etc to see whether they (instead of group) explain differences.

you also have data about the ventricles, and even if this is not your focus, this is a proof of concept as there is solid evidence that SSD and C differ in ventricle volume, you could then express the differences for HPC, amygdala and thalamus relative to the differences in ventricles (easier effect size to understand, e.g., (not validated with your data!!!) the group difference for the HPC was in the same range as that for ventricles ...

alternatively, please show brain images with e.g. the HPC in C and HPC in SSD, a possibility is to use dashed lines to show the borders and thereby help the reader to see the differently sized subcortical structures. Even if Fig 1-3 are a neat way to present it, a table with the medians per group and subcortical region as well as these figures would be helpful. Alternatively, provide the (STATA) code to generate the figure on zenodo too, so that the numbers can be read off (very hard from the figures).

Fig 4-6 are not informative. Firstly, they are not corrected for multiple testing. Secondly, cognitive tests measuring similar abilities should be aggregated before entering in a GLM (regression model). thirdly, the figures imply that you did a regression with that many predictors fir n=10. And this is clearly wrong. 
A correlation plot of HPC, amygdala, thalamus (left and right) and cognitive abilities and then 3 plots: one for group C, one for group SD and one for group SND would be an option, but again - these correlations are based on n=15, 10 and 10 which is massively underpowered.

even if you use all n=35 and do regressions, it is massively underpowered.

the entire cognitive evaluation + brain data is not meaningful with such a small N. These cognitive tests were done to ensure the classification into SND and SD are appropriate. These are not needed to be correlated with the brain volumes.

I recommend to solely focus (brain images, tables) on the brain volume data, reporting it for subnuclei but also ventricles.
report the cognitive evaluations as check for your split into SD and SND

it is also wrong to state that AD is not associated with executive function deficits as you wrote (and AD is not identical with dementia)

and please explain why n=70 in the published data file but here only n=35 - which data got excluded and why.

I do highly appreciate that you have open data, thanks for providing it!

there are a few minor English issues, none that hampers the understanding

minor: page 1 line 46: is twice as much (not twice higher)

page 2 line 8: no consensus (not not consensus)

page 2 line 11: to have the largest (not more)

page 2 line 50: positive (not possitive)

page 3 line 3: by a neuro... (not for neuropsychiatrist)

page 3 line 9: Raw (not row)

page 3 line 11: carried out the screening, i.e. cognitive evaluations

page 7 line 7: expect (not expecting)

Round 2

Reviewer 1 Report

The authors address all the reviewer's comments carefully and have improved their manuscript considerably. I have no further comments. 

Author Response

We acknowledge the reviewer for the comments and suggestions for improving our paper.

Reviewer 2 Report

Dear authors,

thanks for the revision of your manuscript and clarifying many of my points. 
Please note that it should be persons / patients with a schizophrenia spectrum diagnosis and not schizophrenic patients. But I leave this up to the reader (and editor) as it does not hamper understanding. 

Minor issue 

you corrected "The age was ..." to "The age were ..."

but it was correct as "The age was ..."
(age is singular, you can say "ages were ..." but this is not appropriate in this context as you refer to age as a variable / predictor

Author Response

We acknowledge the reviewer for the comments and suggestions for improving our paper.

1. thanks for the revision of your manuscript and clarifying many of my points. 
Please note that it should be persons / patients with a schizophrenia spectrum diagnosis and not schizophrenic patients. But I leave this up to the reader (and editor) as it does not hamper understanding. 

Thank you for your comment. We will have the expression "schizophrenic patients".

2. you corrected "The age was ..." to "The age were ..."

but it was correct as "The age was ..."
(age is singular, you can say "ages were ..." but this is not appropriate in this context as you refer to age as a variable / predictor

Thank you for your suggestion. The text has been changed.